# Risk factor of elevated matrix metalloproteinase-3 gene expression in synovial fluid in knee osteoarthritis women

Delmi Sulastri[1][☉]*, Arnadi Arnadi[2][☉], Afriwardi Afriwardi[3][‡], Desmawati Desmawati[1][☉], Arni Amir[3][‡], Nuzulia Irawati[4][‡], Amel Yanis[5][‡], Yusrawati Yusrawati[6][‡]

1 Faculty of Medicine, Department of Nutrition, Universitas Andalas, Padang, West Sumatera, Indonesia,
2 Faculty of Medicine, Department of Orthopedics and Traumatology, Riau University, Pekanbaru, Indonesia,
3 Faculty of Medicine, Department of Physiology, Universitas Andalas, Padang, West Sumatera, Indonesia,
4 Faculty of Medicine, Department of Biomedicine, Universitas Andalas, Padang, West Sumatera, Indonesia,
5 Faculty of Medicine, Department of Psychiatry, Universitas Andalas, Padang, West Sumatera, Indonesia,
6 Department of Obstetrics and Gynecology, Universitas Andalas, Padang, West Sumatera, Indonesia

☉ These authors contributed equally to this work.
‡ These authors also contributed equally to this work
* delmisulastri@med.unand.ac.id

**Data Availability Statement:** All relevant data are within the manuscript and its Supporting information files.

## Abstract

### Introduction

Metalloproteinases-3 (MMP3) are the main enzymes involved in cartilage degradation. Several genetic and non-genetic factors can increase the expression of MMP3 in patients with osteoarthritis (OA). This study aims to analyze the risk factors associated with the expression of the MMP3 gene rs679620 fluid synovial knee OA patients.

### Methods

A cross-sectional study was conducted at the orthopedic polyclinic Arifin Achmad Riau Province and Ibn Sina Hospital in Pekanbaru City. Ninety women who experienced knee OA were taken as samples by consecutive sampling and then signed the informed consent. Data were obtained through interviews using a questionnaire about characteristics, followed by weight and height measurements. Interleukin-1 β (IL-1β) and Tumor Necrosis Factor (TNF-α) were examined from the synovial fluid using the enzyme-linked immunosorbent assay (ELISA) method. The Metalloproteinases-3 (MMP3) gene polymorphism rs679620 was obtained from the DNA analysis of joint fluid results in the Biomedical Laboratory of the Faculty of Medicine, Andalas University. The data was processed computerized and then analyzed using the correlation Spearman-Rank, and chi-square tests. The results of statistical analysis are considered significant if the p-value is 0.05.

### Results

The MMP3 rs679620 gene polymorphism of the mutant type was 88.9%, with the same proportion of AG and GG alleles (44.4%). Subjects aged ≥ 60 years were 53.3%, 85.6% did not work and 84.4% had menopause. The highest degree of OA was grade 2 (53.3%), most of

**Funding:** This study was funded by the medical faculty of Andalas University and LP2M Andalas University (Grant no 21/UN.16.02/Fd/PT.01.03/2021) in collecting and data analysis. The funders had no role in study design, data collection, and analysis, the decision to publish, or preparation of the manuscript.

**Competing interests:** The authors have declared that no competing interests exist.

whom had a risky nutritional status (84.4%). The median expression of the MMP3 rs679620 gene was 5.28 copies number. There is a significant relationship between MMP3 gene polymorphism rs679620, age, IL-1β, and TNF-α with MMP3 gene expression rs679620. There is no significant relationship between BMI, work status, and menopausal status with MMP3 gene expression rs679620. Conclusion. MMP3 gene polymorphism rs679620, age, levels of IL-1β and TNF-α are risk factors for increased MMP3 gene rs679620 expression in female knee OA.

## Introduction

Osteoarthritis (OA) is the most common form of arthritis. Some people call this degenerative joint disease or "wear and tear" arthritis. Joint cartilage begins to break down, with changes also affecting the underlying bone. These changes occur slowly and get worse over time. Osteoarthritis causes pain, stiffness, and swelling. In some cases, it can be accompanied by decreased function and cause disability, making it difficult for sufferers to carry out daily tasks or work [1].

Osteoarthritis is no longer a degenerative disease, but age is still a risk factor. The World Health Organization (WHO) estimates that by 2025 the elderly population will increase by 414% compared to 1990. Along with increasing life expectancy, the prevalence of OA also increases [2]. The Framingham study reported the radiographic prevalence of knee OA in adults aged 45 was 19.2% and 27.8% in the Osteoarthritis Project in the Johnston Region [1]. The results of the third National Health and Nutrition Examination Survey (NHANES III) showed that about 37% of participants aged > 60 years had knee OA [3].

The radiological prevalence of knee OA in Indonesia has reached 15.5% in men and 12.7% in women between the ages of 40–60 years, 5% at the age <40 years, 30% at the age 40–60 years, and 65% at age >61 years [4]. According to Basic Health Research 2018, the prevalence of joint disease based on the diagnosis of health workers in Indonesia is 11.9% and, based on symptoms, 24.7% [5,6]. It is estimated that 40% of the population over 70 years suffer from OA, and 80% have limited mobility in various degrees, from mild to severe, which results in reducing the quality of life of sufferers [7].

Therapies that can cure OA have yet to be found. Osteoarthritis can be prevented, and even symptoms can be reduced by increasing understanding of the disease's pathogenesis and pathophysiology so that the risk factors can be modified. Metalloproteinases (MMPs) are the main enzymes involved in cartilage degradation [8,9]. The MMP3 enzyme degrades the extracellular matrix and activates other serine proteases that influence the bone degradation process. Matrix metalloproteinase-3 (MMP3) is the main degradative enzyme that plays the most role in cartilage destruction besides MMP1, MMP9, and MMP13 and is the most powerful in the incidence of OA [10,11].

Several studies have proven differences in MMP3 expression at various stages of OA [9]. Chen et al. conducted a study on 90 people with OA with an average age of 58.3 years, finding differences in MMP3 expression between OA patients and normal people where grade III OA had a higher MMP3 expression than grades 1 and 2 [12]. Georgiev et al. reported the same results in 56 OA patients and 31 controls. This study found that the median serum MMP-3 was significantly higher in OA patients than in controls, with the mean MMP3 levels in general OA being higher than in knee OA [13].

The MMP3 gene is one of the genes associated with the incidence of OA. This gene is the gene encoding the MMP3 protein synthesis. Several SNPs of MMP3 were associated with the

incidence of OA, but several countries report different SNPs. A case-control study of 100 male OA patients and 197 healthy men, with a mean age of 51.18 ± 7.849 years in the control group and 63.35 ± 5,786 years in the OA case group, reported the MMP3 Gene SNP polymorphism rs639752, rs520540, rs602128, and rs679620 were associated with an increased risk of OA in men in northern China [7,14]. Different results were reported by Tong et al. in 431 female participants (200 cases and 231 controls) at Hong Hui Hospital, Xi'an Jiaotong University School of Medicine from 2015 to 2016, where the MMP3 gene polymorphisms G minor allele SNP rs650108 and A minor allele SNP rs715572 is a risk factor for OA [15]. Different results were also reported by Murat Kara et al. in 100 Turks suffering from OA and 83 healthy people, where there was no association between MMP-1 SNP rs5854 (A/G) and MMP-3 SNP rs679620 gene polymorphisms [16].

Biomechanical changes during OA will stimulate chondrocyte cells to produce several inflammatory mediators, such as interleukin-1beta (IL-1β) and Tumor Necrosis Factor-alpha (TNF-α) in tissues and joint fluid of OA patients [17]. Increased synthesis of these cytokines will inhibit compensatory pathways of chondrocyte synthesis, which is required to restore the integrity of the degraded extracellular matrix (ECM) and upregulate MMP-3 in synovium and human chondrocytes. In addition, this inflammatory reaction also triggers increased MMP3 synthesis [18]. Activated synoviocytes, mononuclear cells, or articular cartilage produce cytokines. Its affectsmetalloproteinase (MMP) gene expression [19,20]. Receptor levels augment the catabolic effect of IL-1β in this disease. IL-1β and TNF-α) significantly upregulate MMP-3 from human synovium and chondrocytes. Neutralization of IL-1β and/or TNF-α). Upregulation of MMP gene expression could be a way for logical development in potential medical therapy for OA [20]. IL-1β and TNF-α, the main cytokines produced by synovium and chondrocytes, play an essential role in cartilage degradation. Interleukin-1β (IL-1β) is a significant marker early in the OA process. It increases cartilage extracellular matrix damage, collagenolytic metalloprotease activity, and nitric oxide (NO) activity. It also induces chondrocyte apoptosis. Meanwhile, IL-1β will induce changes in cartilage homeostasis and reduce growth factorsactivityy [20,21].

The difference in the level of development and severity of OA is still a challenge in the development of science. Non-genetic factors such as nutritional status, type of occupation, age, and menopausal status significantly contribute to the initiation and development of OA. Being overweight and obese are risk factors that can trigger and worsen OA. Several studies have consistently shown an association between being overweight and obese with knee OA. The first National Health and Nutrition Examination (NHANES I) reported that obese women had a nearly four times greater risk of knee OA than non-obese women, while obese men had an almost five times higher risk. Individuals who are overweight in their thirties are at greater risk of developing OA later in life [22]. Obesity is a significant risk factor for OA because obesity is thought to explain the risk of developing OA through increased wear and tear on weight-bearing joints. Half of a person's body weight rests on the knee joint during walking. Increased body weight will increase the burden on the knee joint. Losing 5 kg of body weight in women can reduce the risk of symptomatic knee osteoarthritis by 50% [23].

Adipose tissue is a metabolic and endocrine organ active in obese patients. This tissue releases adipokines and cytokines such as interleukin-1β (IL-1β) and tumor necrosis factor-α (TNF-α), which have the potential to cause systemic inflammation due to excess body fat. Several adipokines mediate inflammatory effects and have catabolic roles in the pathophysiology of OA. The role of adipokines as a significant factor in the pathogenesis of OA has been widely reported. Adipokines have proinflammatory and catabolic effects on cartilage. It increased catabolic enzyme production, especially matrix metalloproteinase-3 (MMP3) [24].

Until now, the interaction between risk factors for OA is not well understood, so it is necessary to identify high-risk individuals to improve the prevention and treatment process by

regulating MMP3 expression. This study aims to analyze the risk factors associated the expression of the MMP3 gene rs679620 fluid synovial knee OA patients.

## Materials and methods

Observational research with a cross-sectional study design was conducted on women diagnosed with knee OA who were treated at the orthopedic outpatient clinic, Arifin Achmad Hospital, and Ibnu Sina Hospital, Pekanbaru City. A total of 90 women with knee OA were taken as samples by consecutive sampling according to the inclusion and exclusion criteria, then signed the informed consent. The informed consent sheet contains an explanation of the research, namely the purpose, method of sampling, benefits for the subject, and a statement that participation is voluntary. If the respondent refuses, there will be no impact on the health services they receive.

Primary data collection begins with obtaining approval from the research ethics committee (Ethical Clearance), then population screening is carried out. The diagnosis of knee OA was established based on the American College of Rheumatology (ACR) Subcommittee criteria obtained through history taking, clinical examination, and knee radiographs [25].

The characteristics were obtained through interviews using a questionnaire, and BMI was determined by directly measuring the weight and height of the research subjects. Body weight was measured using a Seca digital scale with an accuracy of 0.06–0.01 kg, while body height was measured using a Stanley Malo microtoise with an accuracy of 0.1 cm. Interviews were conducted by researchers assisted by five trained personnel; from interviews obtained, data on characteristics (age, education, occupation), menopausal status, employment status, and physical activity. All research samples were taken from the synovial fluid of the knee joint as much as 3–5 mL; before that, the skin at the sampling site was first disinfected using alcohol. Synovial fluid was stored in a vacutainer with a temperature of -200 C, then sent to the Biomedical Laboratory of the Faculty of Medicine, Universitas Andalas, to examine MMP3 gene polymorphisms, levels of I,L1β, TNF-α and MMP3 gene expression using RT-PCR. The stages of laboratory examination are:

1. DNA Isolation (Genomic DNA Purification) DNA was isolated from all synovial fluid samples using the GeneJET Genomic DNA Purification Kit-ThermoFisher (Work procedure according to the kit's instruction manual). Purified Whole Gold Mag-Mini Blood Genome DNA and stored at -80˚C after centrifugation. Sequenom MassARRAY Assay Design 3.0 software (Sequenom, Inc, San Diego, CA, USA) was used to design the multiplexed Mass EXTEND SNP assay. Genotyping was performed using the Sequenom MassARRAY RS1000 (Sequenom, Inc.) according to the manufacturer's protocol. Sequenom Typer 4.0 Software™ (Sequenom, Inc.) was used to analyze the data. The MMP3 gene was isolated using the MMP3 807 Forward (5'–3') primer AGAAATGCAGAAGTTCCTTGG reverse (5'–3') GGCCAAAATGAAGAGATCAA. Primers corresponding to SNPs. Test the success and effectiveness of the insulation were controlled using the standard electrophoresis technique using 1.5% agarose with a voltage of 100 volts for 45 minutes. The isolated DNA results were then used as a template for in vitro reactions at the stage of examining the target polymorphism of the MMP3 gene. The position of the MMP3 gene polymorphism was identified by sequencing methods and RFLP-PCR techniques. The primer used in the sequencing activity is one of the primers used in the in vitro amplification stage [7].

2. Creation of cDNA
   It made cDNA using RevertAid First Strand cDNA Synthesis Kit (Fermentation). This kit uses RevertAid M-MUL V reverse transcriptase, which has lower RNAse activity and is

**Table 1. Real-time PCR stages.**

| Early denaturation | 95˚C for 10 seconds |
|---|---|
| Target DNA amplification Denaturation Annealing Extension | 95˚C for 10 seconds optimization (600 C) for 10 seconds, 72˚C for 10 seconds |

equipped with an RNAse inhibitor rib block that protects RNA from RNAse activity and temperatures over 550C. GADPH was used as a control for cDNA formation. cDNA was prepared by mixing template (RNA), reaction buffer, rib block, and reverse transcriptase M-MuL V incubated at 420C for 60 minutes and followed by termination at 700C for 5 minutes. The cDNA formed can be stored at -200 C for two to three weeks or -800 C for the long term.

3. Determination of MMP3 Gene Expression rs679620
Genes were amplified with LightCycler® TaqMan DNA Master Mix kit (Roche) using a Real-timePCR machine (LightCycler® 2.0-Time PCR System Real, Roche), primer concentrations between 0.2–1 M for forward and reverse, respectively. A probe is 0.2–0.4 M. Data analysis was carried out in three stages, namely (Table 1):
Gene expression analysis using real-time PCR can be done with two approaches, relative and absolute quantification. Relative quantification can be done again with two models, namely the Relative Standard Curve Method and the Comparative Cycle Threshold (CT) Method (ΔΔCt). The CT method is used if the efficiency of the analyzed gene is not much different and does not require a normal curve as the analysis using the Relative Standard Curve Method. This study analyzed the expression of the MMP3 SNP gene rs679620 by absolute quantification and Comparative CT Method (ΔΔCt).

4. Measurement of IL-1β and TNF-α levels
Interleukin-1β (IL-1β) and TNF-α were examined from the synovial fluid using the enzyme-linked immunosorbent assay (ELISA) method with manual ELISA from Bioeureaux and IL-1β reagent from Quantikine. While, TNF-α was examined using the Human TNF-α kit Elisa from eBioscience.

5. Statistical analysis
all data were processed using a computer and displayed in the form of a distribution of characteristics (mean age, education level, type of occupation), MMP3 gene polymorphisms, mean MMP3 expression, IL1β levels, TNF-α levels, menopausal status, and BMI. To see the relationship between the two variables, used chi-square statistical test and Spearman-Rank correlation and the results of statistical analysis are considered significant if the p-value is 0.05. Research approval was obtained from the health and research ethics committee, Faculty of Medicine, University of Riau No.B/147/UN19.5.1.18/UEPKK/2021.

# Results

## Demographic data

Pekanbaru City is the capital and largest city in Riau Province. Geographically, the city is flanked by Siak Regency in the north and east, Kampar Regency in the north, south, and west, and Pelalawan Regency in the south and east. The area of Pekanbaru City is 632.26 km2 or 0.71 percent of the total area of Riau Province. The results of the 2016 Central Statistics Agency survey, Pekanbaru City has a population of 1,064,566, consisting of 546,400 male residents and

518,166 female residents, with a population growth rate of 0.0255%. This research was conducted in two major hospitals in Pekanbaru City, namely the orthopedic poly Hospital Arifin Achmad Riau Province and Ibnu Sina Hospital. The sample obtained based on the inclusion criteria was 100 people. After being screened by taking into account the exclusion criteria and completeness of the data, it was found that ten respondents had extreme values, so the number of samples that could be analyzed was 90 people. Distribution of data consisting of characteristic data (age, education level, type of work, employment status, menopausal status) and data on the degree of OA and nutritional status.

Matrix metalloproteinase-3 is a member of the MMP family, which is the main enzyme that plays a major role in the destruction of joint cartilage. Several factors can affect MMP-3 levels in joint synovial fluid. Table 2 shows the distribution of risk factors associated with the expression of the MMP3 s679620 gene in the study subjects.

In Table 2, It can be seen that the study subjects aged ≥ 60 years were more (53.3%), highly educated 55.6%, most of them did not work (85.6%) with 84.4% had risky nutritional status and menopausal. MMP3 gene polymorphism rs679620 mutant type 88.9%, MMP3 gene allele

**Table 2. Frequency distribution of research subjects based on risk factors of MMP3 gene expression rs 679620 (n = 90).**

| Risk Factors | n | % |
|---|---|---|
| Age | | |
| < 60 years | 42 | 46,7 |
| ≥ 60 years | 48 | 53,3 |
| MMP3 Gene rs679620 Polymorphism | | |
| WildType | 10 | 11,1 |
| Mutant | 80 | 88,9 |
| MP3 Gene rs679620 Polymorphism | | |
| AA (Wild Type) | 10 | 11,1 |
| AG (Heterozigot) | 40 | 44,4 |
| GG (Homozigot) | 40 | 44,4 |
| Education Level | | |
| Low | 28 | 31,1 |
| Moderate | 12 | 13,3 |
| Hight | 50 | 55,6 |
| Occupational Status | | |
| Doesn't work | 77 | 85,6 |
| Working | 13 | 14,4 |
| Menopause Status | | |
| Not Menopause yet | 14 | 15.6 |
| Menopause | 76 | 84.4 |
| BMI | | |
| No Risk (Normal) | 14 | 15.6 |
| at risk (Obesity +overweight | 76 | 84.4 |
| OA Degree | | |
| Degree 2 | 49 | 54.4 |
| Degree 3 | 36 | 40.0 |
| Degree 4 | 5 | 5.6 |
| MMP3 Gene Expression | | |
| Under-Expression | 45 | 50 |
| Over-Expression | 45 | 50 |

**Table 3. Mean age, BMI, IL-1β, TNF- and median MMP3.** Gene Expression rs679620 on Research Subject.

| Variable | Mean±SD | Median | Min | Maks. |
|---|---|---|---|---|
| Age (years) | 60,66 ± 9,95 | - | 40 | 87 |
| BMI | 27,35 ± 4,06 | - | 20,40 | 47,10 |
| IL-1β (pg/mL) | 431,51±192,69 | - | 140,20 | 1247,06 |
| TNF-α (ng/L) | 106,03±48,82 | - | 31,07 | 309,09 |
| MMP3 Gene Expression (*Copies Number*) | - | 5,28* | 0,16 | 108,33 |

* Data distribution is not normal.

rs679620 with the most AG and GG alleles 44.4% respectively. The highest degree of OA. was grade 2 (54.4%). Subjects with over-expression and under-expression of MMP3 in the same amount (50%:50%).

The mean age, BMI, IL-1β, TNF-α, and the median expression of the MMP3 gene rs679620 can be seen in Table 3. The mean age of the subjects was 60.66 ± 9.95 years, with the youngest age 40 years and the oldest 87 years. The mean BMI, IL-1β, and TNF-α were 431.51±190.84 pg/mL and 105.03±48.82 ng/L, respectively. MMP3 gene expression data is not normally distributed, so it is presented in the form of median = 5.28 Copies Number. The risk factors associated with the expression of joint synovial fluid MMP3 can be seen in the following table.

In Table 4, Subjects with mutant type polymorphism, 53.8 have over-expression of the MMP3 gene and xxx overexperience occurs in the AG alel (62.5%). Over-expression occurs mostly at age $\geq$ 60 years (62.5%), BMI is not at risk (57.1%), Working (53.8) and has menopause (52.6). There was a significant relationship between the MMP3 gene polymorphism

**Table 4. Risk factors associated with MMP3 gene expression.**

| Variable | MMP3 Gene Expression | | | | Total | | p value |
|---|---|---|---|---|---|---|---|
| | Under-expression | | Over-Expression | | | | |
| | n | % | n | % | n | % | |
| MMP3 Gene Polymorphism | | | | | | | |
| WildType | 8 | 80 | 2 | 20 | 10 | 100 | 0,04* |
| Mutant | 37 | 46,3 | 43 | 53,8 | 80 | 100 | |
| MMP3 Gene Alel | | | | | | | |
| AA (Wild Type) | 8 | 80 | 2 | 20 | 10 | 100 | 0.03* |
| AG (Heterozigot) | 15 | 37.5 | 25 | 62.5 | 40 | 100 | |
| GG (Homozigot) | 22 | 55 | 18 | 45 | 40 | 100 | |
| Age | | | | | | | |
| < 60 years | 27 | 64.3 | 15 | 35.7 | 42 | 100 | 0.01* |
| $\geq$ 60 years | 18 | 37.5 | 30 | 62.5 | 48 | 100 | |
| BMI | | | | | | | |
| No Risk | 6 | 42.9 | 8 | 57.1 | 14 | 100 | 0.38 |
| At Risk | 39 | 51.3 | 37 | 48.7 | 76 | 100 | |
| Occupational Status | | | | | | | |
| Doesn't work | 39 | 50.6 | 38 | 49.4 | 77 | 100 | 0.50 |
| Working | 6 | 46.2 | 7 | 53.8 | 13 | 100 | |
| Menopause Status | | | | | | | |
| Not Menopause yet | 9 | 64.3 | 5 | 35.7 | 14 | 100 | 0.19 |
| Menopause | 36 | 47.4 | 40 | 52.6 | 76 | 100 | |

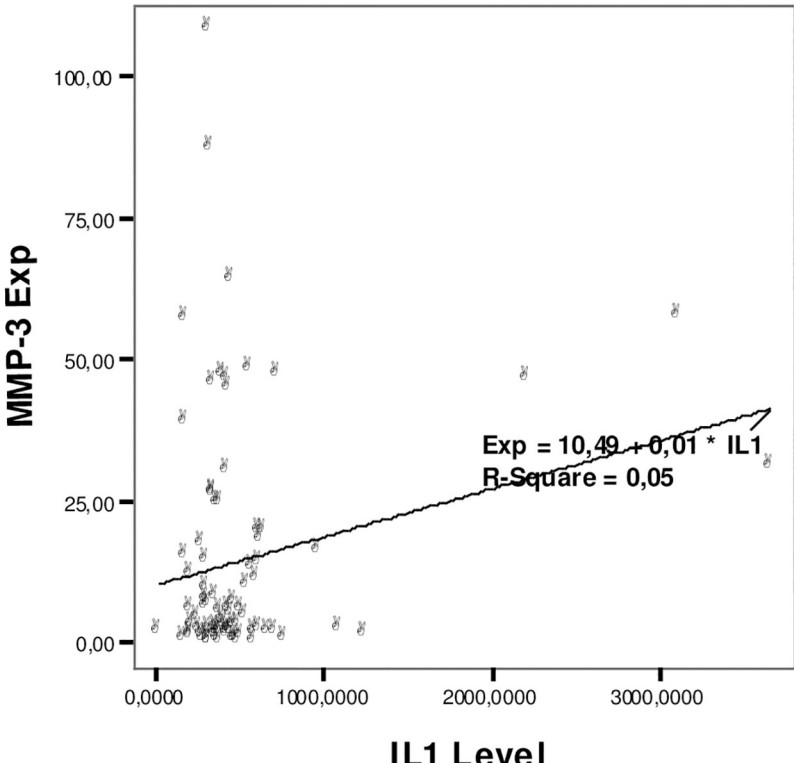

**Fig 1. Relationship of IL1β levels with MMP3 gene expression rs679620.**

rs679620 and age with the expression of the MMP3 gene rs679620 ($p < 0.05$), and there was no relationship between nutritional status, occupation, menopausal status, and the expression of the MMP3 gene rs679620 ($p > 0.05$).

The relationship between IL1β and TNF-α levels with the expression of the MMP3 rs679620 gene can be seen in Figs 1 and 2.

Fig 1 shows that the higher the level of IL-1β, the higher the expression of the MMP3 rs679620 gene. There was no relationship between IL1β-α levels and the expression of the synovial fluid MMP3 gene ($p = 0.05$).

Fig 2 shows the higher the TNF-α level, the higher the expression of the MMP3 rs679620 gene. There was a significant relationship between TNF-α levels and the expression of the MMP3 gene in synovial fluid with moderate strength and a positive pattern ($p = 0.01$, $r = 0.277$).

## Discussion

### Age with MMP 3 gene expression rs679620

Age is one of the risk factors associated with the incidence of OA. Biological changes that occur during the aging process will cause thinning of joint cartilage, weakened muscle strength, poor proprioception, and higher oxidative stress. The older a person is, the more cumulative exposure to risk factors will be [2]. This study found that the mean age of the research subjects was 60.66 ± 9.95 years, more were aged ≥ 60 years (53%), and 62.5% over-expression of the MMP3 gene was found at age ≥ 60 years. There was a significant relationship

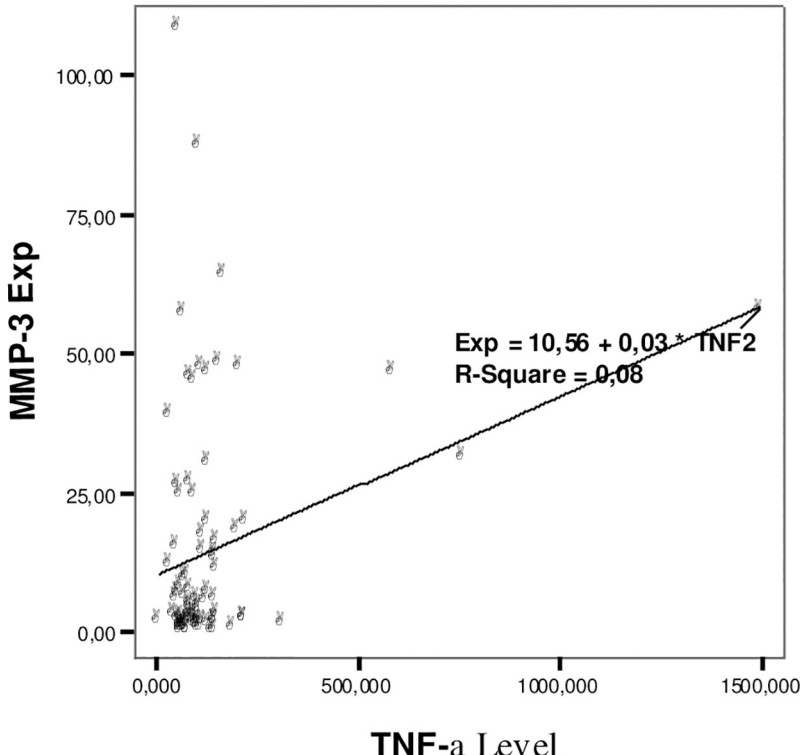

**Fig 2. The relationship between TNF-α levels and the expression of the MMP3 rs679620 gene.**

between age and the expression of the MMP3 gene rs679620 (p = 0.01). The results of this study are in line with research conducted by Liu et al. (2018) in 58 postmenopausal female knee OA patients, with a mean age of 61.35 ± 4.82 years, and this result is higher than the average age of OA patients in Guo et al.'s study (2017), which is 51.18±7.849 years. This difference may be due to differences in study design, where Guo et al.'s case-control study was conducted on 100 male OA patients. At the age of over 50 years, in elderly patients with knee OA, joint flexibility will decrease [7,26,27].

Komosinska-Vassev also reported similar results on 81 healthy people aged 6–62. The study by Komosinska-Vassev et al. reported that serum MMP-3 levels were positively correlated with age (r = 0.44, p = 0.00001) [28]. The same results were also reported by Fujita et al.,2011 in mice with intervertebral discs. It was found that MMP-3 mRNA expression and the ability of MMP-3 expression to respond to TNF-α stimulation decreased significantly with age [29].

Several mechanisms underlying the role of aging in the development of osteoarthritis are reduced muscle mass and increased fat mass, which results in increased joint load. This process will increase the production of adipokines and cytokines, resulting in low-grade systemic inflammation and extracellular matrix changes, including the accumulation of advanced glycation end products. Inflammatory reactions will cause damage to the bone matrix in Osteoarthritis. These cytokines (IL-1β, TNF α) can inhibit the production of proteoglycans and type II collagen and stimulate chondrocytes to release degenerative enzymes (proteases) that damage, among others, MMP3. Aggrecan is the second major component after collagen type 2 in the extracellular matrix of joint cartilage. Aggrecan interacts with hyaluronic acid to form large proteoglycan aggregates. Proteoglycan is part of the extracellular matrix component, which is most responsible as a framework that resists shear and stress forces on the joint cartilage

surface and functions to form synovial fluid to maintain the elasticity of the cartilage substance. If the amount of proteoglycan is reduced, the water content will decrease, and the breakdown of joint collagen will occur. This alters the mechanical properties of cartilage and makes it more susceptible to degeneration, disruption of the extracellular matrix, and decreased cell density in the meniscus and ligaments. The decreased subchondral bone function can alter joint mechanics due to reduced numbers of osteocytes and ligaments. Changes in mineral composition, mitochondrial dysfunction, increased oxidative stress, and reduced autophagy in chondrocytes promote catabolic processes and cell death [9,30].

## Polymorphism with MMP 3 gene expression rs679620

SNP rs679620, located in exon 2 of the MMP-3 gene, results in an amino acid change (Glu4Lys) at residue 45. Based on the Hapmap (haplotype map) in the August 2010 stage IIþIII database, this SNP is the only SNP common across the region encoding the gene MMP-3, which causes amino acid changes in the Central Europe (CEU) population (Utah residents of Northern and Western European descent). These amino acid changes have the potential to alter their interactions with other amino acids in this region and influence MMP-3 activation. Beyzade et al. reported that the SNP rs679620 could be used as a marker to determine MMP-3 levels [31]. Several studies have proven this relationship, although it is still controversial. A case-control study in 100 male OA patients and 197 healthy men, with a mean age of $51.18 \pm 7.849$ years in the control group and $63.35 \pm 5.786$ years in the OA case group, reported that the MMP3 gene polymorphism rs679620 was associated with increased risk of OA in men in northern China [7,14].

In this study, it was found that 88.90% of the MM3 rs679620 gene had mutations, with the most alleles being AG and G G (44.4%). This result is higher than that reported by Chen et al. in 2012, where 74% of 423 respondents had mutations, with the most allele being AG (53.4%). Similar results were also reported by Silva et al. in 2012 on 104 patients at the Faculty of Dentistry, University of Pittsburgh. With periapical lesions. A total of 80.77% of respondents had a mutant MMP3 rs 679620 gene, with 60.57% being the AG allele. There was a significant relationship between the MMP3 gene rs679620 with the incidence of caries and periapical lesions (p = 0.004). MMP3 enzymes play a role in bone remodeling, immune response, caries, and tooth development [32].

The results of this study are higher than the research conducted by Kara et al. (2016) in the Turkish population; Kara et al. obtained the MMP3 rs679620 gene that experienced mutations was 59%, and there was no relationship between the MMP3 rs679620 gene and the incidence of OA [16]. The MMP3 gene rs679620 is a missense mutation that changes the function of the wildtype gene and results in the conversion of Lysine to Glutamine in the final protein [32]. The difference between the results of this study and several other research results may be due to the ethnic differences of the study respondents where our study was conducted on OA patients in Pekanbaru City with a predominantly Malay ethnicity, while Chen et al. conducted a study on Caucasian residents, in North Staffordshire, England, and Cara in Turkish population. Ethnicity is a complex thing consisting of genetic makeup, social construction, cultural identity, and behavioral patterns. Ethnic classification systems are often used to explore and predict the health problems of different populations. Individuals from different ethnic backgrounds vary in genetic makeup, behavior, comorbidities, immune profiles, and disease risk [33]. Research conducted by Cruz-Almeida et al. (2014) reported that there was a relationship between ethnicity/race of knee OA patients with clinical symptoms and extremity function [33]. The Cohort Study conducted by Malaysian Elders Longitudinal Research (MELoR) on residents aged > 55 years reported that there was a significant difference in the incidence of

OA between ethnic Malays 44.6%, ethnic Chinese 23.5% and ethnic Indians 31.9%, and Ethnic Malays have the highest prevalence of OA [34].

MMP-3 protein is mainly secreted by synovial cells, fibroblasts, and cartilage cells and in small amounts by osteoclasts. When the MMP enzyme is activated, it will directly degrade proteoglycans in the extracellular matrix. The activated MMP-3 enzyme then activates zymogens from other MMPs, which causes the degradation of extracellular matrix components such as fiber adhesive proteins and laminin. This causes MMP degradation products, namely mucopolysaccharides, to be released into the joints simultaneously and cause joint inflammation. Serum MMP-3 levels reflect inflammation, and MMP-3 gene polymorphisms are associated with disease activity levels over time, with the highest level of disease activity occurring at SNP rs679620 [12].

This study showed a significant relationship between MMP3 expression based on the MMP-3 gene polymorphism rs679620 ($p < 0.05$). The same result was also reported by Chen et al., 2012 in the SNP analysis, that there was a significant relationship between circulating MMP-3 levels and MMP-3 SNP rs679620. The G allele of rs679620 had the highest circulating levels of MMP-3. Chen et al. also reported that a study conducted on 90 OA patients showed significant differences in MMP 3 in OA patients with the control group, and MMP-3 levels were positively correlated with OA. Similar results were also reported by Georgiev et al. in 56 OA patients and 31 controls. This study found that the median serum MMP-3 was significantly higher in OA patients than in controls, with the mean MMP3 levels in knee OA higher than in controls (13). mRNA for MMP-3 is expressed more intensely in cartilage, with a biphasic pattern in early and late-stage disease, most prominently in late-stage conditions. Data suggest that gene expression for MMP-3 is differentially regulated in human articular chondrocytes and, in individual cells, is related to the depth of chondrocytes beneath the cartilage surface and the nature and extent of cartilage lesions [35].

### Levels of IL-1β and TNF-α with MMP3 gene expression rs679620

Interleukin-1 beta (IL-1β) and TNF-α are key cytokines involved in the pathogenesis of OA. Interleukin-1 beta is one of 11 representatives of the IL-1 (IL-1β) family. Their synthesis in the joint is regulated by chondrocytes, osteoblasts, cells that make up the synovial membrane, and mononuclear cells that are previously present in the joint or infiltrate its structures during the inflammatory response. Patients with OA have elevated levels of IL-1β and TNF-α both in the synovial fluid, synovial membrane, cartilage, and subchondral bone layer [20]. These cytokines play an important role in promoting synovial membrane inflammation and osteophyte formation. TNF-α inhibits the synthesis of the main extracellular matrix (ECM), components: proteoglycans, type II collagen and cartilage connective protein (HAPLN1). In addition, IL-1β and TNF-a also increase the expression and release of various cartilage-damaging MMPs including MMP-3, and have the effect of exacerbating inflammation by inducing the production of proinflammatory cytokines [36]. the higher the level of IL-1β, the higher the expression of MMP3. There is no correlation between levels of IL-1β and the expression of the MMP3 gene rs679620, while for TNF-α it is found that the higher the TNF-α, the higher the expression of the MMP3 gene rs679620. There is a positive correlation with moderate strength between TNF-α and MMP3 expression. These results support the existing theory, where IL-1β and TNF-α can increase MMP3 gene expression. Metalloproteinase enzymes can cause cartilage damage, mediated by tumor necrosis factor (TNF-α) and interleukin (IL)-1β. Proinflammatory cytokines, including TNF-α, IL-1β, are responsible for extracellular matrix regulation, cartilage degradation, and chondromic apoptosis. it. The same result was also reported by

Afifah et al (2019) where human chondrocyte cell ls was induced by IL-1β causing an increase in synovial fluid MMp3 expression [37].

The insignificant increase between IL-1β and MMP3 expression is thought to be because all respondents are poly orthopedic patients who routinely visit the hospital and have received anti-inflammatory drugs. Non-steroidal anti-inflammatory drugs (NSAIDs) are the most widely used drug therapy in OA. Non-steroidal anti-inflammatory drugs (NSAIDs) are known to be effective in relieving symptoms and reducing inflammation. Several investigators have reported that administration of anti-inflammatory drugs can inhibit IL-1β-induced MMP-3 expression and activity and markedly suppress nuclear translocation of NF-κB by blocking IkB-degradation. alpha in human chondrocytes. A study conducted by Efstathiou et al (2017) on 36 OA patients who were given NSIDs reported that MMP-3 levels were significantly reduced after NSAID treatment [38]. Matrine is an anti-inflammatory, also shown to inhibit IL-1β-induced matrix metalloproteinase expression by suppressing MAPK and NF-B activation in human chondrocytes in vitro. NF-kB plays an important role in cartilage degradation in O.A. Under unstimulated conditions, the NF-κB dimer is located in an inactive form in the cytoplasm bound to the IκB molecule. In OA, chondrocytes express various NF-B-mediated cytokines and chemokines, such as TNF-α, IL-1β, IL-6, NF-B ligand receptor activator (RANK) (RANKL) and IL-8, which increase the production of MMPs, decreases collagen and proteoglycan synthesis and acts in a positive feedback loop to increase NF-kB activation. In addition, in human chondrocytes, NF-B inhibitors reduce IL-1β-induced production of MMP-3 and MMP-13 [39].

## BMI with MMP 3 gene expression rs679620

Obesity and overweight have long been recognized as strong risk factors for OA, especially knee OA Results from the Framingham Study showed that women who lost about 5 kg of body weight had a 50% reduction in the risk of knee OA symptoms. The same study also found that weight loss was strongly associated with a reduced risk of knee OA, and has been shown to reduce pain and disability in knee OA [40].

This study found that 84.4% of respondents had nutritional status at risk, and 51.3% of subjects with nutritional status at risk had under-expression of the MMP3 gene. There was no significant relationship between BMI at risk and the expression of the MMP3 gene (p = 0.38). Risk factors for overweight and obesity in OA can cause mechanical stress on the knee joint. There are about 0.7 MPa loads on the joints when standing, walking between 5 and 10 MPa, and exercising more than 18 MPa. Some studies have shown that the production of Extra Cellular Matrix (ECM) by chondrocytes is susceptible to various mechanical signals mediated by this load. Moderate exercise benefits cartilage formation, whereas excessive stress or static stress disrupts anabolic and catabolic homeostasis in cartilage [41].

The increase in fat mass due to aging is associated with an increase in the number of adipocytes and proinflammatory macrophages in adipose tissue, and these cells produce several cytokines and adipokines that can induce the onset of OA. However, the mechanism of action of these mediators in OA remains unclear. Obesity and increased fat mass can also cause meta-inflammation metabolic changes. Meta-inflammatory reactions are associated with increased levels of circulating free fatty acids, hyperglycemia, and oxidative stress, which can negatively impact joint tissue by promoting matrix destruction [9].

This result is contrary to the existing theory. It can be caused by respondents having criteria for at-risk nutritional status. Respondents are also patients who routinely seek treatment at the hospital orthopedic clinic and have received anti-inflammatory drugs so that the impact of obesity on MMP3 expression has been reduced. Decrease. Historically, the high incidence of

knee OA in obese people has led to the assumption that mechanical loads such as excessive body weight can cause joint wear and tear but that OA also occurs in non-weight-bearing joints, such as the hands and wrists. This shows that an unreasonable expected load cannot fully explain the relationship between OA and obesity [42].

## Menopause status with MMP 3 gene expression rs679620

Postmenopausal Osteoarthritis (OA) is a joint disease that most often occurs in people over 45 years with a high disability rate and dramatically affects patients' health and quality of life. The decrease in the hormone estrogen in this condition causes a failure of osteoblast activity, decreased intestinal calcium absorption, calcium reabsorption in bone, and increased urinary calcium excretion resulting in a prolonged calcium imbalance. In addition, estrogen can provide genomic and non-genomic effects on various cells, including female chondrocytes. Estrogens can also affect articular cartilage homeostasis by modulating the expression of molecules associated with cartilage damage, such as different growth factors, inflammatory cytokines, matrix metalloproteinases, and reactive oxygen species [27]. Estrogen and progesterone receptors have been identified in human articular cartilage, synovium, and ligaments, making these tissues susceptible to hormonal changes, including menopause [27,43].

In this study, it was found that 84.4% of the research respondents had menopause. The results of the interaction test between menopausal status and the expression of the MMP3 gene rs679620 showed that postmenopausal subjects had more over-expression (52.6%) and there was no significant relationship between menopausal status (p = 19). The results of this study are different from the results reported by Rollick et al (2018) in experimental animals, it is found that aging and menopause affect gene expression patterns of knee joint tissue differently. A study conducted on 59 OA patients in 2016 also reported that a lack of estradiol was associated with the pathogenesis of postmenopausal OA, the inflammatory factor IL-1β and TNF-α were elevated in the serum and synovial fluid of OA patients. We did not find any studies examining the interaction between menopausal status and the expression of the MMP3 rs 679620 gene. However, based on research conducted by Rollick and Liu, it has been shown that menopausal status is related to gene expression. The results of our study cannot prove this, presumably because, in general, the respondents are already menopausal, so the respondents are almost homogeneous [27,44].

The decrease in the hormone estrogen will reduce the rate of lipoprotein synthesis in the liver and intestines by affecting the lipoprotein lipase enzyme. It will affect lipid metabolism, reduce free fatty acid flux, decrease fatty acid oxidation and increase the incorporation of fatty acids into triglycerides. Increased fat synthesis (lipogenesis) will cause an increase in Fat Mass (FM) in premenopausal women. This situation will trigger weight gain and obesity. Estrogen also increases the activity of osteoblasts and suppresses the action of osteoclasts. It is an indirect cause of the incidence of OA because it causes the knee joint burden to increase [43,45].

## Occupational status with MMP 3 gene expression rs679620

Occupation is one of the risk factors for OA in women. This study found that most of the research subjects had jobs as housewives (75.6%), with the frequency and intensity of force used in the medium category. There is no relationship between the type of work with MMP3 Gene Expression rs679620 (p>0.05). The results of this study contradict the theory, which states that jobs that use knee strength are at high risk for suffering from knee OA. This condition is often found in heavy physical workers, especially those who use a lot of knee strength to support body weight. The prevalence of knee OA was higher among port workers, farmers, and miners than administrative workers [42]. The most common occupational risk factors for

knee OA are heavy physical workload, frequent exposure to biomechanical stressors such as bending the knee, kneeling or squatting, standing for long hours ($\geq$ 2 hours per day), walking 3 km/day, climbing stairs, lifting weight ($\geq$ 10 kg), jump, and vibration. A study in the UK showed that knee OA was five times more common in workers aged 55 years who were exposed to several risk factors, such as heavy lifting (>25 kg) by kneeling/squatting or climbing stairs. Those who performed regular knee flexion without lifting weights had only a 2.5 times greater risk of developing OA (46). The sample of this study mostly had jobs as housewives, with light to moderate levels of activity, and did not use the knees as a place to support body weight [46].

## Conclusions

Female knee OA patient has MMP3 gene polymorphism rs679620 Mutant type by 88.9% with the most alleles being AG and GG. 53.8% of mutant type MMP3 Gene Polymorphism was over-expressed, with the most over-expression found in the GA allele (62,5%). There is a relationship between age and MMP3 gene polymorphism rs679620 with synovial fluid MMP3 expression in female knee OA patients. There is a correlation between the levels of IL1β and TNF- with the expression of MMP3 rs679620 in synovial fluid in female knee OA patients. There is no relationship between OA degree, BMI, occupation, and menopausal status with synovial fluid MMP3 expression in female knee OA patients.

## Supporting information

**S1 Data.**
(PDF)

## Acknowledgments

We are grateful to all respondents who participated in this study.

## Author Contributions

**Conceptualization:** Delmi Sulastri, Arnadi Arnadi, Nuzulia Irawati, Yusrawati Yusrawati.

**Data curation:** Delmi Sulastri, Desmawati Desmawati, Amel Yanis.

**Funding acquisition:** Amel Yanis.

**Investigation:** Arnadi Arnadi, Amel Yanis.

**Methodology:** Delmi Sulastri, Arnadi Arnadi, Arni Amir, Yusrawati Yusrawati.

**Project administration:** Arnadi Arnadi, Nuzulia Irawati, Amel Yanis.

**Resources:** Afriwardi Afriwardi.

**Software:** Desmawati Desmawati.

**Supervision:** Afriwardi Afriwardi, Arni Amir.

**Validation:** Afriwardi Afriwardi, Desmawati Desmawati, Yusrawati Yusrawati.

**Visualization:** Desmawati Desmawati, Arni Amir, Nuzulia Irawati.

**Writing – original draft:** Afriwardi Afriwardi, Desmawati Desmawati.

**Writing – review & editing:** Delmi Sulastri, Desmawati Desmawati, Nuzulia Irawati, Amel Yanis, Yusrawati Yusrawati.

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
