## [Decision Letter · Decision Letter 0]

7 Feb 2023

PONE-D-22-26715Risk Factor of Elevated Matrix Metalloproteinase-3 (MMP-3)  G ene E xpression in Synovial Fluid In Knee Osteoarthritis WomenPLOS ONE

Dear Dr. Delmi Sulastri,

Thank you for submitting your manuscript to PLOS ONE. After careful consideration, we feel that it has merit but does not fully meet PLOS ONE’s publication criteria as it currently stands. Therefore, we invite you to submit a revised version of the manuscript that addresses the points raised during the review process.

Please submit your revised manuscript by Mar 24 2023 11:59PM. If you will need more time than this to complete your revisions, please reply to this message or contact the journal office at plosone@plos.org. Please include the following items when submitting your revised manuscript:A rebuttal letter that responds to each point raised by the academic editor and reviewer(s). You should upload this letter as a separate file labeled 'Response to Reviewers'.A marked-up copy of your manuscript that highlights changes made to the original version. You should upload this as a separate file labeled 'Revised Manuscript with Track Changes'.An unmarked version of your revised paper without tracked changes. You should upload this as a separate file labeled 'Manuscript'.If applicable, we recommend that you deposit your laboratory protocols in protocols.io to enhance the reproducibility of your results. Protocols.io assigns your protocol its own identifier (DOI) so that it can be cited independently in the future. For instructions see: https://journals.plos.org/plosone/s/submission-guidelines#loc-laboratory-protocols. Additionally, PLOS ONE offers an option for publishing peer-reviewed Lab Protocol articles, which describe protocols hosted on protocols.io. Read more information on sharing protocols at https://plos.org/protocols?utm_medium=editorial-email&utm_source=authorletters&utm_campaign=protocols.

We look forward to receiving your revised manuscript.

Kind regards,

Ewa Tomaszewska, DVM Ph.D

Academic Editor

PLOS ONE

Journal Requirements:

- file:///D:/C%20DRIVE/Downloads/oamjms-10b-1319%20(2).pdf

In your revision ensure you cite all your sources (including your own works), and quote or rephrase any duplicated text outside the methods section. Further consideration is dependent on these concerns being addressed.

Yes. This study funded by medical faculty of Andalas University and LP2M Andalas University (Grant no 21/UN.16.02/Fd/PT.01.03/2021) in collecting  and data analysis

We are grateful to medical faculty of Andalas University and LP2M Andalas University for funding this research project (Grant no 21/UN.16.02/Fd/PT.01.03/2021)

However, funding information should not appear in the Acknowledgments section or other areas of your manuscript. We will only publish funding information present in the Funding Statement section of the online submission form. 

Yes. This study funded by medical faculty of Andalas University and LP2M Andalas University (Grant no 21/UN.16.02/Fd/PT.01.03/2021) in collecting  and data analysis

Reviewers' comments:

Reviewer's Responses to Questions

**Comments to the Author**

1. Is the manuscript technically sound, and do the data support the conclusions?

Reviewer #1: Yes

2. Has the statistical analysis been performed appropriately and rigorously? 

Reviewer #1: Yes

3. Have the authors made all data underlying the findings in their manuscript fully available?

Reviewer #1: Yes

4. Is the manuscript presented in an intelligible fashion and written in standard English?

Reviewer #1: Yes

5. Review Comments to the Author

Reviewer #1: General comments

This manuscript aims at analyzing the risk factors associated with the expression of the MMP3 gene rs679620 in knee synovial fluid in osteoarthritis patients. Despite several minor issues detailed below (which, however, need to be addressed), authors manage to fulfill sufficiently their aim.

Minor comments

(Title and elsewhere throughout MS, as well) Please, do not use acronyms in headings;

(Abstract, Introduction) Methods;

(References) 1. … N. Classifications in Brief: Kellgren-Lawrence Classification of Osteoarthritis…

(Introduction, 7th paragraph and elsewhere throughout MS, as well) please, do not start sentences with acronyms;

… growth factors activity...

(10th par) please, check for font size;

(Materials and Methods, 3rd par) The CT method…

(Age with MMP 3 Gene Expression rs679620, 1st-2nd par) please, check for correct in-text referencing;

ref Komosinska-Vassev et al. not in References;

(3rd par) is aggrecan an advanced glycation end product? Please, explain;

(Polymorphism with MMP 3 Gene Expression rs679620, 1st par) please, introduce CEU;

(2nd÷5th par) please, check for correct in-text referencing;

Almeida et al. (2014)≠(30);

(Levels of IL-1β and TNF-α with MMP3 Gene Expression rs679620, 1st-2nd par) please, check for correct in-text referencing;

… in OA. Under…

(Menopause Status with MMP 3 Gene Expression rs679620, 2nd par) please, check for correct in-text referencing.

6. PLOS authors have the option to publish the peer review history of their article (what does this mean?). If published, this will include your full peer review and any attached files.

Reviewer #1: No

---

## [Author Response · Author response to Decision Letter 0]

14 Mar 2023

We have revised this article according to suggestions and input from editors and reviewers.

---

## [Decision Letter · Decision Letter 1]

20 Mar 2023

Risk Factor of Elevated Matrix Metalloproteinase-3 Gene Expression in Synovial Fluid In Knee Osteoarthritis Women

PONE-D-22-26715R1

Dear Dr. Delmi Sulastri,

We’re pleased to inform you that your manuscript has been judged scientifically suitable for publication and will be formally accepted for publication once it meets all outstanding technical requirements.

Kind regards,

Ewa Tomaszewska, DVM Ph.D

Academic Editor

PLOS ONE

Additional Editor Comments (optional):

Reviewers' comments:

Reviewer's Responses to Questions

**Comments to the Author**

1. If the authors have adequately addressed your comments raised in a previous round of review and you feel that this manuscript is now acceptable for publication, you may indicate that here to bypass the “Comments to the Author” section, enter your conflict of interest statement in the “Confidential to Editor” section, and submit your "Accept" recommendation.

Reviewer #1: All comments have been addressed

2. Is the manuscript technically sound, and do the data support the conclusions?

Reviewer #1: Yes

3. Has the statistical analysis been performed appropriately and rigorously? 

Reviewer #1: Yes

4. Have the authors made all data underlying the findings in their manuscript fully available?

Reviewer #1: Yes

5. Is the manuscript presented in an intelligible fashion and written in standard English?

Reviewer #1: Yes

6. Review Comments to the Author

Reviewer #1: General comments

I do not have any further particular concerns to express about the manuscript. Authors addressed sufficiently all points raised by me.

7. PLOS authors have the option to publish the peer review history of their article (what does this mean?). If published, this will include your full peer review and any attached files.

Reviewer #1: No

---

## [Editor Report · Acceptance letter]

23 Mar 2023

PONE-D-22-26715R1 

Risk Factor of Elevated Matrix Metalloproteinase-3 Gene Expression in Synovial Fluid In Knee Osteoarthritis Women 

Dear Dr. Sulastri:

I'm pleased to inform you that your manuscript has been deemed suitable for publication in PLOS ONE. Congratulations! Your manuscript is now with our production department. 

Kind regards, 

on behalf of

Professor Ewa Tomaszewska 

Academic Editor

PLOS ONE